# Low prevalence of *Plasmodium falciparum histidine-rich protein 2* and *3* gene deletions in malaria-endemic states of India

Sri Krishna[1‡], Shrikant Nema[2‡], Ruchika Sangle[2], Amreen Ahmad[1], Akansha Singh[2], Devendra Kumar[2], Anil K. Verma[1], Venkatachalam Udhayakumar[3], Aparup Das[1], Anup R. Anvikar[2], Praveen K. Bharti[2]*

1 ICMR-National Institute of Research in Tribal Health, Jabalpur, Madhya Pradesh, India, 2 ICMR-National Institute of Malaria Research, New Delhi, India, 3 The Task Force for Global Health, Decatur, Georgia, United States of America

‡ SK and SN contributed equally to this work as joint first author.
* saprapbs@yahoo.co.in

## Abstract

Rapid diagnostic tests (RDTs) are crucial for diagnosing malaria in resource-limited settings. These tests, which detect the histidine-rich protein 2 (PfHRP2) and its structural homologue PfHRP3, are specifically designed to identify *Plasmodium falciparum*. Deletion of the *Pfhrp2* gene in parasite has been reported in India and other malaria-endemic countries. Therefore, periodic surveillance of *Pfhrp2* and *Pfhrp3* genetic deletions is crucial. We conducted a study to examine these gene deletions in *P. falciparum* isolates from nine malaria-endemic states in India. In this study, we analyzed 1,558 samples that were microscopically confirmed to be *P. falciparum* positive. We isolated genomic DNA from all the aforementioned samples, followed by PCR amplification of the *Pfhrp2/3* gene. The results showed that the deletion rates for *Pfhrp2* and *Pfhrp3* genes were 0.44% and 1.47%, respectively. These findings indicate that the gene deletions in all nine states are at low level. Despite these low deletion rates, continuous surveillance is crucial to monitor the efficiency of HRP2 based malaria RDTs. It is recommend that conducting large-scale studies which include other endemic states in India to gain a more comprehensive understanding of the prevalence and impact of these gene deletions over time. This ongoing surveillance will ensure that diagnostic strategies remain effective and that any emerging trends in gene deletions are promptly addressed to achieve the malaria control and elimination.

## Introduction

Malaria is at the forefront of the World Health Organization's (WHO) disease elimination programs, given its significant global mortality rates each year. In 2022, India contributed about 65.7% of all malaria cases in the WHO South-East Asia region, with *P. falciparum* accounting for nearly 54% of these cases [1]. Despite a substantial reduction in malaria cases in 2023, there were still 0.22 million reported cases, compared to 214 million cases in 2015 [2]. These cases

**Funding:** The author(s) received no specific funding for this work.

**Competing interests:** The authors have declared that no competing interests exist.

predominantly come from rural parts of India, particularly in areas with limited resources and insufficient medical infrastructure [3]. Malaria diagnosis in these regions relies mainly on microscopy and rapid diagnostic test (RDT) kits, each with its own advantages and limitations [3, 4]. Histidine-rich protein II (HRP2), a protein produced exclusively by *P. falciparum*, enables HRP2 RDTs to exhibit high specificity and sensitivity for detecting this parasite compared to other malaria RDTs [5]. The genes encoding HRP2 and its analogue protein, HRP3 (*Pfhrp2* and *Pfhrp3*), are located in the sub-telomeric regions of Plasmodium chromosomes 8 and 13, respectively [5]. These regions are known for extensive genetic diversity and frequent changes during recombination events [6, 7]. HRP3 shares repeat motifs with HRP2, allowing antibodies against HRP2 to cross-react with HRP3. The failure of RDTs to detect *P. falciparum* infections can often be attributed to the absence of detectable HRP2 antigen levels. This may result from genetic variability and deletions at the *hrp2* and *hrp3* loci, along with other factors such as misinterpretation of RDT results, the prozone effect, and *Pfhrp2/3* gene deletions [5]. Recognizing the significant implications of these false negatives, the WHO has recommended revising testing strategies if the local prevalence of false-negative HRP2 RDTs due to gene deletions reaches 5% [8]. Initial evidence of widespread *Pfhrp2* and *Pfhrp3* gene deletions emerged from studies in Peru [9]. In India, Bharti et al. reported low-level deletions of the hrp2 gene across eight highly endemic states, raising concerns about the reliability of HRP2 RDTs in these regions and underscoring the need for continuous monitoring and surveillance [10]. Given the critical role of accurate diagnosis in malaria control and elimination efforts, this study further investigates the prevalence of *Pfhrp2* and *Pfhrp3* gene deletions in parasites from nine malaria-endemic states in India. The aim is to determine whether there have been changes in the prevalence of these deletions, which could impact the effectiveness of current diagnostic tools. By focusing on these gene deletions, the study seeks to enhance our understanding of the genetic dynamics of *Pfhrp2* in India and improve diagnostic accuracy.

## Methods

### Study site and sample collection

The present study utilized stored blood samples collected by the ICMR-NIRTH in 2014, 2017 2019, and 2020 as part of an Therapeutic efficacy study (TES) [11, 12] (Table 2). A total of 1558 samples (microscopically confirmed *P. falciparum*) were collected from the nine malaria endemic states of India (Fig 1), were used for the detection of *Pfhrp2* and *Pfhrp3* gene deletion. The study was approved by ICMR-NIRTH Institutional Ethics Committee to utilise the archived samples in the current study (NIRTH/IEC/01/3162/2020).

### Genomic DNA extraction

Whole blood samples were obtained from patients. Subsequently, 200 µl of blood was utilized for isolation, and DNA extraction was performed using a Qiagen DNA Blood Mini Kit (Qiagen-51306, Hilden, Germany) as per the manufacturer's protocol. The DNA sample was stored at -80˚C for long term storage, and at -20˚C during the molecular biology analyses.

### Amplification of histidine-rich protein 2 and 3 (hrp2/3)

Two step PCR amplification (primary and nested) method was used to amplify a segment encompassing exon 2 of *Pfhrp2* and *Pfhrp3* gene to carried out the gene deletions prevalence, PCR product of *Pfhrp2* and *Pfhrp3* gene resulted in a 222 bp and 216 bp respectively. Sample with *Pfhrp2* and *Pfhrp3* negative results were confirmed using 18S rRNA, *msp1* and *msp2* markers. For the primary PCR, 5 µL of genomic DNA was used as the template, while for the

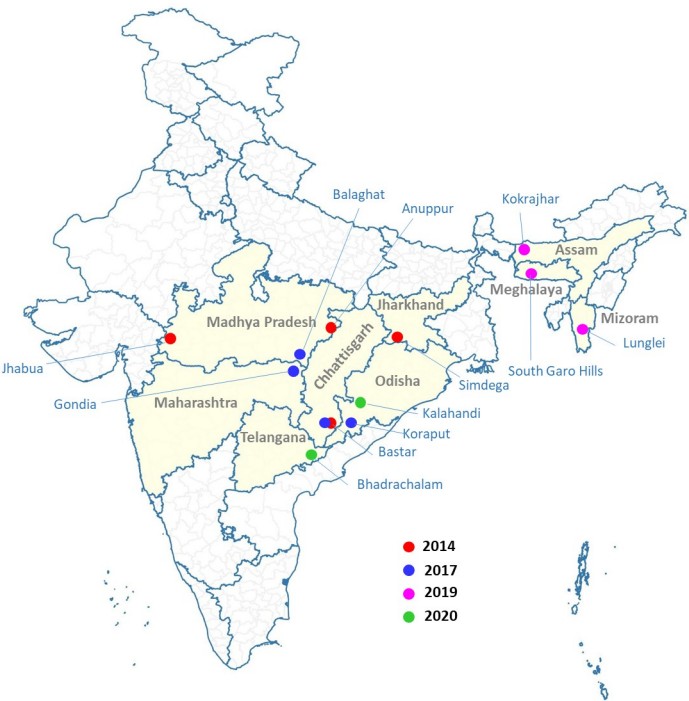

**Fig 1. Map of India highlighting the nine malaria-endemic states.**

nested PCR, 2 μL of a 1:10 diluted primary PCR product served as the template. The PCR reaction was conducted in a 25 μL mixture containing 10X buffer, 1 mM MgCl₂, 0.2 mM of each dNTP, 0.4 μM of each primer, and 0.2 units of Taq polymerase (Invitrogen, Life Technologies). All PCR products were analyzed on a 1.2% agarose gel, and images were captured using a Gel-Doc-It² imager. The details regarding primers and PCR cycling parameters employed for the amplification of *Pfhrp2* and *Pfhrp3* are described in Table 1.

## Amplification of 18s rRNA, merozoite surface protein 1 and 2 (msp1 and msp2)

To assess DNA integrity, samples that were not amplified for the HRP2/3 gene were tested for 18S rRNA, *msp1*, and *msp2* to confirm the presence of genuine HRP2/3 gene deletions. Details regarding the primers and PCR cycling parameters used for amplifying 18S rRNA, *msp1*, and *msp2* are provided in Table 1. The PCR amplified gel images are shown in Fig 2.

### Data analysis

The results of the *Pfhrp2/3* gene deletion were recorded in a Microsoft Excel sheet to estimate the prevalence of gene deletions by state.

## Results

### Gene deletion in *Pfhrp2*

The overall rate of *Pfhrp2* deletion in nine malaria-endemic states is 0.44% (7 out of 1,558). Among these states, Chhattisgarh has the highest level of *Pfhrp2* deletion at 1.1% (4 out of 370), followed by Odisha at 0.7% (1 out of 141), Jharkhand at 0.6% (1 out of 166), and Madhya

**Table 1. Details of primers used in the study.**

| Primer name | Primer sequence | Denaturation | Annealing | Elongation | No of cycles |
|---|---|---|---|---|---|
| Pfhrp2 (exon 2) primary | GGTTTCCTTCTCAAAAAATAAAG TCTACATGTGCTTGAGTTTCG | 94 °C, 1 min | 58°C, 45 sec | 72 °C, 1 min | 35 |
| Pfhrp2 (exon 2) nested | GTATTATCCGCTGCCGTTTTTGCC CTACACAAGTTATTATTAAATGCGGAA | 94 °C, 1 min | 63°C, 45 sec | 72 °C, 1 min | 25 |
| Pfhrp3 (exon 2) primary | GGTTTCCTTCTCAAAAAATAAAA CCTGCATGTGCTTGACTTTA | 94 °C, 45 sec | 53 °C, 1 min | 72 °C, 1 min | 30 |
| Pfhrp3 (exon 2) nested | ATATTATCGCTGCCGTTTTTGCT CTAAACAAGTTATTGTTAAATTCGGAG | 94 °C, 45 sec | 62 °C, 1 min | 72 °C, 1 min | 25 |
| MSP1 (primary) | CTAGAAGCTTTAGAAGATGCAGTATTG CTTAAATAGTATTCTAATTCAAGTGGATCA | 94°C, 1 min | 61°C, 2 min | 72°C, 1 min | 25 |
| MSP1 (Nested) | AAATGAAGGAACAAGTGGAACAGCTGTTAC ATCTGAAGGATTTGTACGTCTTGAATTACC | 94°C, 1 min | 61°C, 2 min | 72°C, 1 min | 30 |
| MSP2 (primary) | ATGAAGGTAATTAAAACATTGTCTATTATA CTTTGTTACCATCGGTACATTCTT | 94°C, 1 min | 61°C, 1 min | 72°C, 1 min | 25 |
| MSP2 (Nested) | AGAAGTATGGCAGAAAGTAAKCCTYCTACT GATTGTAATTCGGGGGATTCAGTTTGTTCG | 94°C, 1 min | 61°C, 1 min | 72°C, 1 min | 30 |
| 18s rRNA Plasmodium (Genus) | TCAAAGATTAAGCCATGCAAGTGA CCTGTTGTTGCCTTAAACTTC | 95°C, 30 sec | 54°C, 45 sec | 72°C, 1 min 30 sec | 35 |
| P. falciparum (nested) | TTAAACTGGTTTGGGAAAACCAAATATATT ACACAATGAACTCAATCATGACTACCCGTC | 95°C, 30 sec | 58°C, 1 min | 72°C, 1 min | 30 |

Pradesh at 0.2% (1 out of 470). No *Pfhrp2* deletion cases were found in Maharashtra, Assam, Meghalaya, Mizoram, and Telangana. In 2020, *Pfhrp2* deletion was reported at a higher rate of 0.77% (1 out of 129), followed by 0.52% (3 out of 569) in 2017, and 0.47% (3 out of 627) in 2014 (Table 2). Interestingly, no cases of deletion were found in 2019. In order to assess the

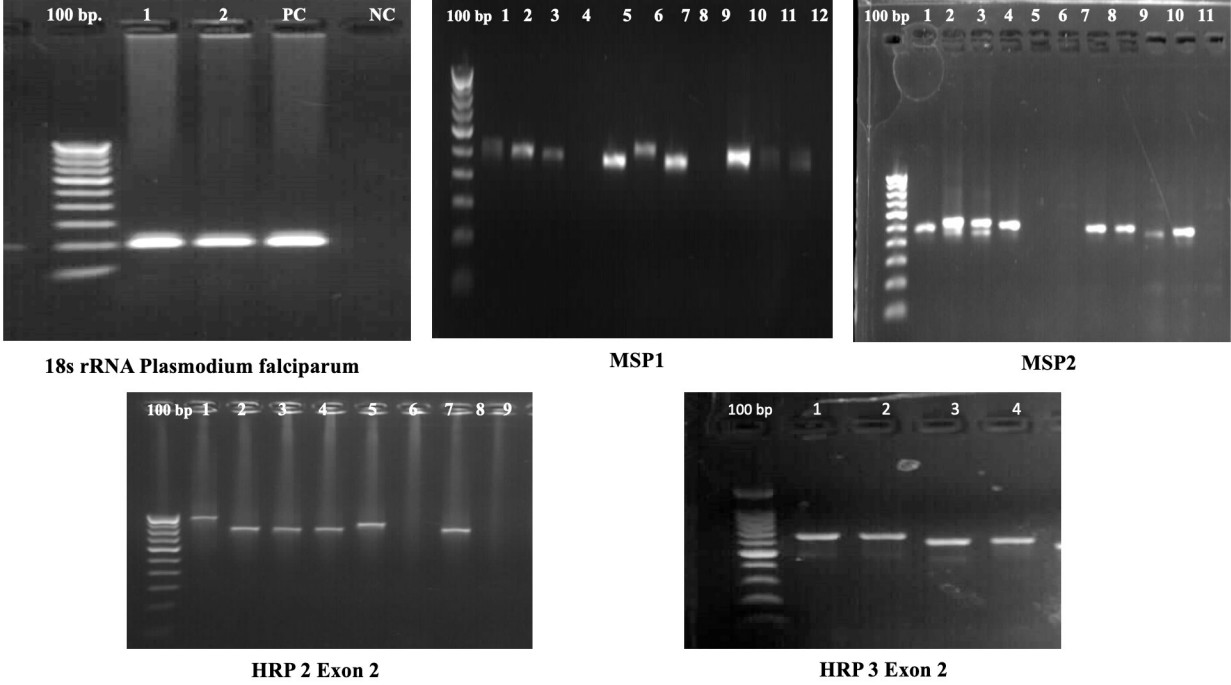

**Fig 2. PCR gel images showing the amplification of hrp2/3, 18s rRNA, msp1, msp2 genes.**

**Table 2. Distribution of hrp2/3 deletion cases across Indian states/year wise within the total sample population.**

| Parameters | Total no. of samples (N) | *hrp2* deletion | *hrp3* deletion | *hrp2* deleted % | *hrp3* deleted % |
|---|---|---|---|---|---|
| 2014 | 627 | 3 | 7 | 0.47 | 1.11 |
| 2017 | 569 | 3 | 11 | 0.52 | 0.70 |
| 2019 | 233 | 0 | 3 | 0 | 0.19 |
| 2020 | 129 | 1 | 2 | 0.77 | 0.12 |
| Madhya Pradesh | 470 | 1 | 3 | 0.2 | 0.6 |
| Chhattisgarh | 370 | 4 | 9 | 1.1 | 2.4 |
| Jharkhand | 166 | 1 | 2 | 0.6 | 1.2 |
| Maharashtra | 118 | 0 | 2 | 0.0 | 1.7 |
| Odisha | 141 | 1 | 4 | 0.7 | 2.8 |
| Assam | 74 | 0 | 1 | 0.0 | 1.4 |
| Meghalaya | 60 | 0 | 1 | 0.0 | 1.7 |
| Mizoram | 99 | 0 | 1 | 0.0 | 1.0 |
| Telangana | 60 | 0 | 0 | 0.0 | 0 |
| **Total** | **1558** | **7** | **23** | **0.44** | **1.47** |

DNA quality of the deleted sample showing *Pfhrp2* deletion, *msp1*, and *msp2* marker genes were amplified, as recommended for evaluating DNA integrity. Interestingly, both marker genes showed successful amplification in this particular sample, while repeated attempts also did not amplify the *Pfhrp2* gene, confirming the *Pfhrp2* deletion (Figs 2 and 3).

## Gene deletion in *Pfhrp3*

The overall rate of *Pfhrp3* deletion in nine malaria-endemic states is 1.47% (23 out of 1,558). Among these states, Odisha has the highest level of hrp3 deletion at 2.8% (4 out of 141),

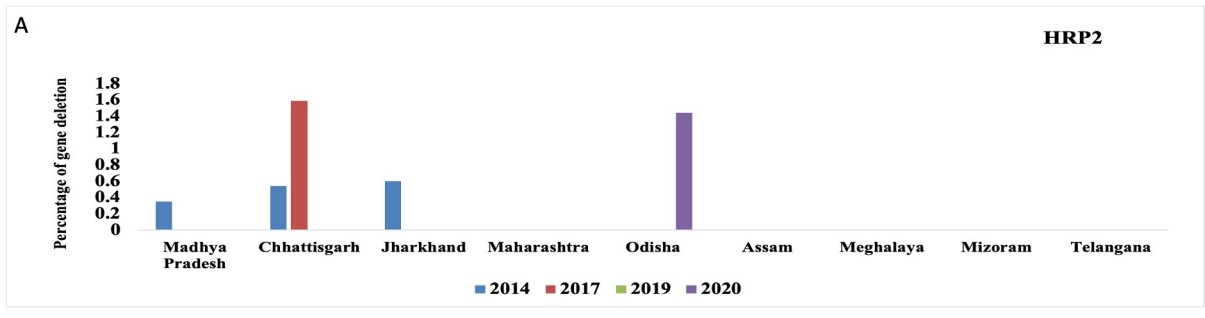

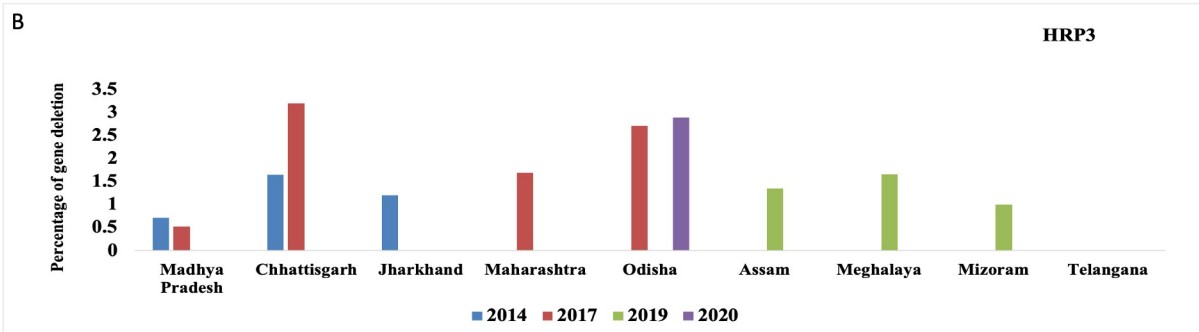

**Fig 3. Year-wise distribution of (A) hrp2 gene deletions and (B) hrp3 gene deletions across Indian states.**

followed by Chhattisgarh at 2.4% (9 out of 370). Meghalaya (1 out of 60) and Maharashtra (2 out of 118) each have a deletion rate of 1.7%, Assam has 1.4% (1 out of 74), Jharkhand has 1.2% (2 out of 166), and Madhya Pradesh has 0.6% (3 out of 470). No *Pfhrp3* deletion cases were found in Telangana. In 2014, *Pfhrp3* deletion was reported at the highest rate of 1.11%, followed by 0.70% in 2017, 0.19% in 2019, and 0.12% in 2020. Population genetic marker genes (*msp1*, and *msp2*) showed successful amplification in these samples, while repeated attempts to amplify the *Pfhrp3* gene failed to amplify the gene confirming *Pfhrp3* deletion (Figs 2 and 3).

## Dual deletion of *Pfhrp2* and *Pfhrp3* genes

Dual deletion of both *Pfhrp2* and *Pfhrp3* genes was found in Madhya Pradesh at 0.21% (1 out of 470), followed by Chhattisgarh at 1.08% (4 out of 370), Jharkhand at 0.60% (1 out of 166), and Odisha at 0.70% (1 out of 141). However, no dual gene deletions were found in Assam, Meghalaya, Mizoram, Maharashtra, and Telangana.

## Discussion

Malaria diagnosis is vital for timely treatment and transmission prevention. In India, RDTs are preferred in resource-limited settings. RDTs have delivered 3.9 billion tests globally since 2010 [1]. The states of Madhya Pradesh, Chhattisgarh, Jharkhand, Maharashtra, Odisha, Assam, Meghalaya, Mizoram, and Telangana, account for nearly 65% of malaria cases in 2023 [2]. These states exhibit varying intensities of malaria transmission due to differences in environmental, socioeconomic, and epidemiological factors (Table 3). This study conducted across nine malaria-endemic states in India have identified low levels of *Pfhrp2* deletions (0.44%) in *P. falciparum* populations. Samples were assessed for 'real' *Pfhrp*2/3 deletions, to accurately interpret test results, we used additional markers, such as the 18S ribosomal RNA gene or *msp1*, and *msp2* genes, as controls. The presence of these genes confirms the integrity of the sample and the efficacy of the PCR process. If the 18S or msp1/2 testing is positive while *hrp2* is not amplified, it is more likely indicative of a genuine deletion. Conversely, if all markers fail to amplify, it may point to a procedural error rather than a true genetic absence. During the analyses of *msp1* and *msp2*, multiclonal infections were also observed, as indicated by multiple

**Table 3. Parasite density and transmission intensity by year and state.**

| Year | States | Parasite density/µL Geometric mean (Range) | Transmission setting |
|---|---|---|---|
| 2014 | Jhabua, Madhya Pradesh | 6868 (5755.9–8195.9) | High |
| | Anuppur, Madhya Pradesh | 2518 (2015.3–3146.4) | Moderate |
| | Bastar, Chhattisgarh | 7869 (6430.5–9629.2) | High |
| | Simdega, Jharkhand | 3032 (2086.4–4405.1) | High |
| 2017 | Balaghat, Madhya Pradesh | 5262.6 (1000–99240) | High |
| | Jagdalpur, Chhattisgarh | 14044 (1080–98200) | High |
| | Kilepal, Chhattisgarh | 10015 (1130–88765) | High |
| | Koraput, Odisha | 7380.8 (1053–98746) | High |
| | Gondia, Maharashtra | 6770.8 (1200–92280) | Moderate |
| 2019 | Udalguri, Assam | 3266 (1270–68955) | Low |
| | South Garo hills, Meghalaya | 3567 (1510–91057) | High |
| | Lunglei, Mizoram | 5474 (1020–96000) | High |
| 2020 | Khammam, Telangana | 5580 (1090–28570) | Low |
| | Kalahandi, Odisha | 12150 (2200–67102) | High |

bands in the isolates (data not shown). In a 2013 study in Chhattisgarh, a 4% deletion rate of the *Pfhrp2* gene was reported, alongside partial gene deletions for *Pfhrp2* and *Pfhrp3* [13, 14]. In 2017–18, Nema et al. observed a 3.8% deletion rate of the *Pfhrp2* gene in Chhattisgarh (unpublished). Similarly, findings in Kolkata indicated a 2.17% deletion rate for both genes [15]. However, Odisha showed a higher incidence of *Pfhrp2* deletion at approximately 17% [16]. A systematic review and meta-analysis by kojom et al. shown that pooled prevalence of *Pfhrp2* deletions was 5% in India. For pfhrp3 deletions, the prevalence was 4% in India [17]. While our study suggests that the prevalence of *Pfhrp2* deletion has not substantially increased compared to previous reports, the WHO recommends baseline surveys in countries with documented *Pfhrp2/3* deletions, and neighboring regions. These surveys are crucial to assess if deletion prevalence surpasses the threshold requiring RDT changes. WHO's response plan includes actions such as identifying new biomarkers, enhancing non-HRP2 RDT performance, market forecasting, and bolstering laboratory networks for molecular characterization [8, 18]. Studies are underway to explore alternative biomarkers like hemozoin [19], heme detoxification proteins (HDP) [20], and Glutamate dehydrogenase (GDH) [21] for developing next-generation RDTs. Continuous surveillance, as recommended by WHO, is essential to ensure the reliability of HRP2-based RDTs. Based on current data, HRP2-based RDTs remain suitable for use in these states. One limitation of the study is that we did not perform PfLDH-based RDT detection according to the hrp2 gene deletion estimation protocol. Accurate detection is essential for effective malaria treatment and management, especially in high-transmission areas. This study enhances global knowledge and informs local health policies to reduce the malaria burden and advance elimination efforts.

## Conclusion

HRP2-based RDT kits have proven immensely beneficial in rural and tribal regions of India, where malaria prevalence is notably high. The current study presents genetic diversity data of *Pf*hrp2 and *Pf*hrp3 genes across nine malaria-endemic states of India. Results confirm a low prevalence of gene deletion in these regions. Among microscopically confirmed samples, only 0.44% exhibited *Pfhrp2* deletion, while 1.47% had *Pfhrp3* deletion across the nine states. This data will aid in comprehending the evolutionary mechanisms linked to the emergence and dissemination of *Pfhrp2*/3 deletions.

## Acknowledgments

The authors would like to thank the study participants and field staff. The manuscript was approved by the Publication Screening Committee of ICMR-NIRTH Jabalpur.

## Author Contributions

**Conceptualization:** Sri Krishna, Shrikant Nema, Praveen K. Bharti.

**Formal analysis:** Shrikant Nema.

**Investigation:** Sri Krishna, Shrikant Nema, Ruchika Sangle, Amreen Ahmad, Akansha Singh, Devendra Kumar, Anil K. Verma, Praveen K. Bharti.

**Methodology:** Sri Krishna, Shrikant Nema, Ruchika Sangle, Amreen Ahmad, Devendra Kumar, Praveen K. Bharti.

**Supervision:** Shrikant Nema, Amreen Ahmad, Praveen K. Bharti.

**Validation:** Shrikant Nema.

**Writing – original draft:** Shrikant Nema, Praveen K. Bharti.

**Writing – review & editing:** Sri Krishna, Shrikant Nema, Anil K. Verma, Venkatachalam Udhayakumar, Aparup Das, Anup R. Anvikar, Praveen K. Bharti.

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
