## [Decision Letter · Decision Letter 0]

16 Sep 2024

PONE-D-24-27945Low prevalence of Plasmodium falciparum histidine-rich protein 2 and 3 gene deletions in malaria-endemic states of IndiaPLOS ONE

Dear Dr. Bharti,

Thank you for submitting your manuscript to PLOS ONE. After careful consideration, we feel that it has merit but does not fully meet PLOS ONE’s publication criteria as it currently stands. Therefore, we invite you to submit a revised version of the manuscript that addresses the points raised during the review process. There is a significant request for revision and additional information, in particular from reviewer 1 and 3 who both are experts in this field. Provide a point-to-point response. Submitting a revised version does not guarantee acceptance of your manuscript.

We look forward to receiving your revised manuscript.

Kind regards,

Henk Schallig, Ph.D

Academic Editor

PLOS ONE

Journal Requirements:

2. We note that your Data Availability Statement is currently as follows: [All relevant data are within the manuscript and its supporting information files.]

3. We note that you have referenced (unpublished) on page 6, which has currently not yet been accepted for publication. Please remove this from your References and amend this to state in the body of your manuscript: (ie “Bewick et al. [Unpublished]”) as detailed online in our guide for authors

5. We note that [Figure 1] in your submission contain [map/satellite] images which may be copyrighted. All PLOS content is published under the Creative Commons Attribution License (CC BY 4.0), which means that the manuscript, images, and Supporting Information files will be freely available online, and any third party is permitted to access, download, copy, distribute, and use these materials in any way, even commercially, with proper attribution. For these reasons, we cannot publish previously copyrighted maps or satellite images created using proprietary data, such as Google software (Google Maps, Street View, and Earth). For more information, see our copyright guidelines: http://journals.plos.org/plosone/s/licenses-and-copyright.

6. Please include your tables as part of your main manuscript and remove the individual files. Please note that supplementary tables (should remain/ be uploaded) as separate "supporting information" files

Reviewers' comments:

Reviewer's Responses to Questions

**Comments to the Author**

1. Is the manuscript technically sound, and do the data support the conclusions?

Reviewer #1: Partly

Reviewer #2: Yes

Reviewer #3: No

2. Has the statistical analysis been performed appropriately and rigorously? 

Reviewer #1: N/A

Reviewer #2: Yes

Reviewer #3: N/A

3. Have the authors made all data underlying the findings in their manuscript fully available?

Reviewer #1: No

Reviewer #2: Yes

Reviewer #3: No

4. Is the manuscript presented in an intelligible fashion and written in standard English?

Reviewer #1: Yes

Reviewer #2: Yes

Reviewer #3: Yes

5. Review Comments to the Author

Reviewer #1: This manuscript describes the results of surveillance of pfhrp2 and pfhrp3 deletions in P. falciparum samples from malaria-endemic states in India. The outcomes are highly relevant for the Indian malaria control programme and show that HRP2 RDTs can still be used reliably for diagnosing malaria. The manuscript itself would benefit from a number of adaptations and additions, mainly in the methods and discussion.

Importantly, Table 1 and 2 were not part of the submission and could thus not be reviewed. As such, the data were not fully available at this point.

Major comments

- Methods: please include more information on the patients from whom the samples were collected. How were these patients identified? Were they symptomatic cases presenting at hospitals? Were there certain inclusion/exclusion criteria?

- How sensitive is the nested PCR + gel readout method? Is it possible that the samples marked with a deletion had low parasite DNA and that there was insufficient hrp product to visualize this on gel?

- Results: apart from percentages, it would be good to also present the absolute numbers of samples and deletions per year and state.

- The start of the discussion (until line 142) is more of a review of the malaria situation in India and the used diagnostics, with the content partly overlapping with the introduction. I suggest the authors rewrite this section, so that it reflects the main outcomes of their study

- Can the authors think of possible reasons why the deletion rate in this study was (much) lower than in some previous ones?

- Lines 155-161: This information is not very relevant for the interpretation of this study, especially when considering that deletion rates are still far below the threshold of 5%.

- A critical reflection on the limitations of the study design should be added to the discussion, e.g. the used methodology for detecting deletions, representativeness of sample, etc.

Minor comments

- Line 22: please rephrase this sentence, as the deletions are not found in RDTs (but in the parasite) and have not been widely reported in India so far.

- Line 42: please rephrase this sentence

- Lines 55-58: the issue of deletions is mentioned twice

- Lines 70-73: this part feels a bit redundant in the introduction and would fit better in the discussion

- Line 88: please explain why the exon 2 segment was chosen for detection of gene deletions

- Line 88-98: this section would be easier to understand if the use of nested PCR is mentioned at the start

- Line 90-91: please clarify what is confirmed exactly by testing for the additional markers

- DNA samples had been stored at -20C for up to 9 years. This is quite long, and at -20C there is a risk of DNA degradation. DNA integrity was checked by detection of msp1 and 2 markers, but these are located on different chromosomes (9 and 2). How do the authors assess the risk of DNA degradation of the subtelomeric regions on which pfhrp2 and 3 are found?

- Line 108: contrary to what is stated here, deletions were reported for Assam, Meghalaya and Mizoram in Figure 2

- Line 110: idem as above, Figure 2 reports deletions in 2017. Please check.

- Line 110-111 and 121-122: were these samples also positive for 18s rRNA?

- Line 151-152: I would omit the numbers from Africa, they are less relevant for this study.

- Line 169: please clarify how the study data will aid to comprehend these evolutionary mechanisms

Layout and editing:

- Please italicize Latin organism names, e.g. Plasmodium falciparum (line 21) and P. falciparum (line 24) (please also check the rest of the manuscript)

- Line 79: add “detection of” before “Pfhrp2”

- Line 113: avoid the use of contractions (didn’t)

- Throughout the manuscript, please be consistent in the notation of the HRP2 gene (hrp2/pfhrp2/P. falciparum hrp2)

Reviewer #2: The manuscript is technically sound and the data of the manuscript support the conclusion.

Statistical analysis performed propely

All the data is available in the manuscript,

the manuscript is written in standard English and intelligible fasion

The abstract is written in the clear language to show the manuscript

the conclusion clear and precise, but a bit sronger recommendation is needed.

Reviewer #3: Low prevalence of Plasmodium falciparum histidine-rich protein 2 and 3 gene deletions in malaria-endemic states of India

The authors present a short report on hrp deletions in India across multiple malaria endemic states spanning several years. The sample number analysed is large and the authors find a few suspected deletions. It would have been great to have some more details on the size of the study sites and distances between the sampling locations. The map (figure 1) has no scale, and if I interpret this correctly, the samples were collected from different locations in different years. As there is no details provided in what malaria transmission is like in the different states and whether it is homogeneous across states, not too many conclusions can be drawn.

No RDT data is presented for the samples here – all samples were microscopy positive but in the absence of direct comparison to RDT performance, no testing recommendations can be made.

It would have been beneficial to see the microcopy parasite density data - overall but also for the suspected deletions.

The report would have benefited from more details in the experimental design and methods used especially in hrp testing where essentially, the inability to amplify a gene suggests that the gene is deleted. The authors need to show the data e.g. add an agarose gel, positive and negative controls and number of repetitions, and the LOD of all assays.

Both tables are missing from the copy of the manuscript that I received.

In the discussion there is no mention of multiclonal infections that potentially contain hrp-deleted minority clones.

The discussion could also include a paragraph on methodological difficulties of hrp testing e.g. When is a deletion a “real” deletion and when is it a PCR failure (only mentioning 18s, msp testing in a couple of sentences is not enough)?

In summary, experiments are not described in sufficient detail and conclusions are not presented in an appropriate fashion and therefore it is very difficult to judge whether the data is supported.

Some more detailed comments:

Line 21 + – Plasmodium falciparum and pfhrp2– please use italics when referring to organisms and gene names throughout the manuscript. Also Genus = capital & species = lower case. Gene names all in lower case.

Line 29 – Please keep nomenclature the same e.g. refer to hrp2 gene as either pfhrp2 or hrp2 but use the same nomenclature throughout the manuscript.

Lins 41-42 – “Despite the significant reduction in malaria cases, in recent years, the total of 0.22 million cases reported in 2023” – the sentence lacks meaning, please revise.

Line 76 – Could you please indicate exactly how the whole blood was collected and stored? Seeing as the hrp deletion analysis depends on PCRs where the absence of product means the presence of a deletion, it would be great to know what state the samples were in when they were tested.

Line 85 - Please give your elution volume.

Line 87 and paragraph – It would be great to include more details on the confirmation PCRs as well as the hrp PCRs. It is important to understand the limit of detection of all PCRs, what positive and negative controls were used, and whether the samples were run multiple times etc. You give some more information in the results, but I believe this should already be clearly explained in the methods as part of the experimental design.

Line 106 + - please always give % with associated numbers e.g. line 105!

6. PLOS authors have the option to publish the peer review history of their article (what does this mean?). If published, this will include your full peer review and any attached files.

Reviewer #1: **Yes: **Norbert van Dijk

Reviewer #2: **Yes: **ABEBECH TESFAYE TOLESSA

Reviewer #3: No

---

## [Author Response · Author response to Decision Letter 0]

5 Nov 2024

Response to Reviewers

Reviewer #1: This manuscript describes the results of surveillance of pfhrp2 and pfhrp3 deletions in P. falciparum samples from malaria-endemic states in India. The outcomes are highly relevant for the Indian malaria control programme and show that HRP2 RDTs can still be used reliably for diagnosing malaria. The manuscript itself would benefit from a number of adaptations and additions, mainly in the methods and discussion. 

Response: We sincerely appreciate the reviewer for their insightful comments and constructive feedback. Their observations have significantly enhanced the quality of our work and have provided valuable perspectives that we will incorporate into our research. In the revised version of manuscript, we have new Figure 2 (so total number figure are 3) and new table 3 (so total number of tables are 3). Thank you for your thoughtful engagement with our manuscript.

1. Importantly, Table 1 and 2 were not part of the submission and could thus not be reviewed. As such, the data were not fully available at this point.

Response: We sincerely apologize for any inconvenience. We have submitted all the necessary data, including Tables along with manuscript. Table 1 includes the details of the primer used in the study, and table 2 show the distribution of hrp2/3 deletion cases across Indian states/year wise within the total sample population and Table 3 includes the Parasite density and transmission intensity by year and state. We are now resubmitting this data for your consideration.

Major comments

2. Methods: please include more information on the patients from whom the samples were collected. How were these patients identified? Were they symptomatic cases presenting at hospitals? Were there certain inclusion/exclusion criteria?

Response: The samples used in this study was an part of therapeutic efficacy studies in previous years, where all febrile patients, who visited the selected study sites (Primary Health Centre/Community Health Centre) were screened by microscopy for malaria. As per the protocol (Ref: METHODS FOR SURVEILLANCE OF ANTIMALARIAL DRUG EFFICACY, 2009), we have enrolled the only microscopy confirmed mono-infected P. falciparum in the study. Thin and thick blood smear were prepared and observed under compound microscope using oil immersion lens at 100x magnification. Around 100 thick blood smears fields were observed by the microscopist before declaring a slide negative and parasite density was calculated by counting the number of parasites against 200 leukocytes. After informed consent, about 2 ml blood was collected in EDTA tube from microscopically confirmed mono P. falciparum cases. Initially, the blood samples were stored at -20°C at study sites and subsequently transported to ICMR-National Institute of Research in Tribal Health (ICMR-NIRTH) Jabalpur in liquid Nitrogen for further molecular analysis. 

This information is already mentioned in the previously published paper (Ref 10,11). 

3. How sensitive is the nested PCR + gel readout method? Is it possible that the samples marked with a deletion had low parasite DNA and that there was insufficient hrp product to visualize this on gel?

Response: Nested PCR combined with gel electrophoresis is highly sensitive for targeting the 18S rRNA gene. Its two-round amplification increases yield and specificity, allowing detection of low DNA quantities, often down to a single copy. So, the chances of missing the low parasite count is negligible. Patients with a parasite density greater than 1000 parasites per microliter were eligible for enrolment (Ref: METHODS FOR SURVEILLANCE OF ANTIMALARIAL DRUG EFFICACY, 2009) (Refer Table 3). It is worth to mention that during hrp2 analysis we checked the integrity of the DNA samples using 18s rRNA, msp1 and msp2 markers. Control samples known to be of good quality were utilized to further validate our findings. Additionally, these samples had previously been amplified for antimalarial resistance marker genes (dhfr, dhps, k13, mdr1) in the prior Therapeutic Efficacy Study (reference 10, 11). Thus, concerns regarding low-density parasites may not be applicable. 

4. Results: apart from percentages, it would be good to also present the absolute numbers of samples and deletions per year and state.

Response: Thank you for the suggestion. Given the word limit of the manuscript, we have presented the requested data in Table 2.

5. The start of the discussion (until line 142) is more of a review of the malaria situation in India and the used diagnostics, with the content partly overlapping with the introduction. I suggest the authors rewrite this section, so that it reflects the main outcomes of their study

Response: We appreciate the reviewer’s view and modified the sentence, although we feel it is important to elaborate the context about Indian malaria situation and preferred diagnostics. Moreover, we have taken care of the overlapping content from the introduction to streamline the narrative. (Please refer to line number 134 – 139 in the clean copy of manuscript).

6. Can the authors think of possible reasons why the deletion rate in this study was (much) lower than in some previous ones?

Response: We would like to point out that in previous large multi-state studies overall prevalence of pfhrp2/3 deletion was lower (<5 % (2.4%), Bharti et al. 2016, Ref no 9). However, in some studies conducted in some selected states with limited sample size showed deletion ranging from 2.17% to 3.8%. Overall, the prevalence of pfhrp2 deletion in India has been below the 5% threshold. Given this scenario, our current findings illustrate it is important to conduct large nationwide studies are important to understand the prevalence of pfhrp2 deletion. It is possible that in some selected pockets pfhrp2 deleted parasites could be higher than in some other sites and therefore appropriate sampling frame work is necessary to get a true prevalence. We speculate use of convenient sampling frame work in some studies may have contributed to this bias. Nevertheless, this study and previous studies with larger samples size have consistently pointed out low prevalence of pfhrp2 deletion and this suggests RDT use in India has not resulted in selection of such parasites. 

 The decrease in HRP2 gene deletions in India, particularly in the context of malaria, can be attributed to several factors. Improved surveillance and monitoring systems may have enhanced the detection and management of malaria cases, including those caused by HRP2-deleted strains. Ongoing vector control measures, such as insecticide-treated nets and indoor residual spraying, may also be limiting the spread of HRP2-deleted strains by reducing overall malaria transmission. 

7. Lines 155-161: This information is not very relevant for the interpretation of this study, especially when considering that deletion rates are still far below the threshold of 5%.

Response: We understand the reviewer’s concerns, we have modified the statement. Nonetheless, it is essential to highlight that this aligns with the WHO's target of a 5% threshold, indicating that a policy change is not needed at this time in India. 

8. A critical reflection on the limitations of the study design should be added to the discussion, e.g. the used methodology for detecting deletions, representativeness of sample, etc.

Response: As previously mentioned, these samples were obtained from a prior study, specifically the Therapeutic Efficacy Study (TES). According to the TES protocol, only Plasmodium falciparum mono-infected samples confirmed by microscopy should be collected, which is why RDTs were not used for these samples. Consequently, we could not strictly follow the protocol for assessing HRP2/3 gene deletion (Ref: Protocol for estimating the prevalence of pfhrp2/pfhrp3 gene deletions among symptomatic falciparum patients with false-negative RDT results, World Health Organization). We have acknowledged that PfLDH-based detection was not performed, and this limitation has now been incorporated as you recommended. (Please refer to line number 166 -167 in clean version of manuscript).

Minor comments

9. Line 22: please rephrase this sentence, as the deletions are not found in RDTs (but in the parasite) and have not been widely reported in India so far.

Response: Thank you for highlighting this important correction. We have addressed the necessary correction. (Please refer to line number 22 in clean version of manuscript).

10. Line 42: please rephrase this sentence

Response: The sentence has been rephrased. (Please refer to line number 42-43 in clean version of manuscript).

11. Lines 55-58: the issue of deletions is mentioned twice

Response: We have addressed the necessary correction.

12. Lines 70-73: this part feels a bit redundant in the introduction and would fit better in the discussion.

Response: Thank you for your suggestion. Following your recommendation, we have moved lines 70-73 from the introduction section to the discussion section at the end. (Please refer to line number 168-171 in clean version of manuscript).

13. Line 88: please explain why the exon 2 segment was chosen for detection of gene deletions

Response: Exon 2 encodes a significant portion of the HRP2 protein, which is essential for the malaria parasite's ability to evade the host's immune system and is a key target in rapid diagnostic tests (RDTs). Deletions in this region can lead to the loss of HRP2 expression, impacting the reliability of these diagnostic assays. 

14. Line 88-98: this section would be easier to understand if the use of nested PCR is mentioned at the start.

Response: We have revised the sentence to place greater emphasis on nested PCR, providing a clearer picture for the audience. (Please refer to line number 84-85 in clean version of manuscript).

15. Line 90-91: please clarify what is confirmed exactly by testing for the additional markers. 

Response: The msp1 and msp2 genes are recommended for confirmation after hrp2/3 gene deletion because they are crucial for accurately detecting malaria parasites. hrp2/3 is commonly used in rapid diagnostic tests, but some strains may undergo deletion of this gene, leading to false negatives. By targeting msp1 and msp2, markers to know the presence of P. falciparum parasite even in cases where hrp2/3 is absent and rule out the produceorial error. This dual approach enhances diagnostic accuracy. (Please refer to line number 95-99 in clean version of manuscript).

16. DNA samples had been stored at -20C for up to 9 years. This is quite long, and at -20C there is a risk of DNA degradation. DNA integrity was checked by detection of msp1 and 2 markers, but these are located on different chromosomes (9 and 2). How do the authors assess the risk of DNA degradation of the subtelomeric regions on which pfhrp2 and 3 are found?

Response: As mentioned, these samples were collected for TES studies in previous years. As per the protocol, parasite density of these samples were more than or equal to 1000 parasite/ul. Moreover, these samples were previously amplified for the drug resistance marker (dhfr, dhps, mdr, k13). Furthermore, we need to clarify that we have stored the blood samples were stored at -20C at study sites. Once the samples were transported to ICMR-NIRTH, Jabalpur, they were Stored at -80C. (Please refer to line number 81-82 in clean version of manuscript).

While we initially checked the integrity of the DNA samples using msp1 and msp2 markers. The targeted amplification allowed us to determine the presence of intact sequences. Control samples known were utilized to further validate our findings. Through these approaches, we have effectively evaluated the integrity of the pfhrp2 and pfhrp3 regions, despite the potential risks associated with long-term storage at -80°C.

17. Line 108: contrary to what is stated here, deletions were reported for Assam, Meghalaya and Mizoram in Figure 2

Response:. We have revised Figure 3 (previously figure 2) and included in the updated version of manuscript.

18. Line 110: idem as above, Figure 2 reports deletions in 2017. Please check.

Response: Figure 3 (Previously Figure 2) has been corrected. 

19. Line 110-111 and 121-122: were these samples also positive for 18s rRNA?

Response: Yes, the samples that were hrp3 negative were also confirmed positive for 18S rRNA.

20. Line 151-152: I would omit the numbers from Africa, they are less relevant for this study.

Response: That’s correct. We appreciate your suggestion, and we have removed the number from Africa accordingly. (Please refer to line number 154-155 in clean version of manuscript).

21. Line 169: please clarify how the study data will aid to comprehend these evolutionary mechanisms

Response: This types of study help us to determine the evolutionary mechanisms, such as selective pressure from the routine use of RDTs, have contributed to the selection and spread of pfhrp2-deleted parasites. Fortunately, the low incidence of deletions means that we do not need to conduct detailed molecular studies to assess whether pfhrp2-deleted parasites share a common evolutionary origin due to RDT selection pressure. 

Layout and editing:

22. Please italicize Latin organism names, e.g. Plasmodium falciparum (line 21) and P. falciparum (line 24) (please also check the rest of the manuscript)

Response: We have italicized the names of organisms throughout the manuscript.

23. Line 79: add “detection of” before “Pfhrp2”

Response: As per your suggestion, we have added “detection of.”

24. Line 113: avoid the use of contractions (didn’t)

Response: We have rephrased the contraction.

25. Throughout the manuscript, please be consistent in the notation of the HRP2 gene (hrp2/pfhrp2/P. falciparum hrp2)

Response: Throughout the manuscript we have mentioned the gene name in italicized. For example, Pfhrp2. But for protein names, typically not italicized and often written with the capitalized letter. For example, HRP2.

Reviewer #2: The manuscript is technically sound and the data of the manuscript support the conclusion.

Statistical analysis performed properly

All the data is available in the manuscript,

the manuscript is written in standard English and intelligible fashion

The abstract is written in the clear language to show the manuscript

the conclusion clear and precise, but a bit stronger recommendation is needed.

Response: Thank you for your feedback. We appreciate your comments on the manuscript. We are glad to hear that the statistical analysis was performed properly and that the manuscript is presented in standard English. We will work on strengthening the recommendations in the conclusion to enhance its impact. Your insights are valuable, and we will ensure that these revisions improve the overall quality of the manuscript. Thank you again for your constructive comments!

Reviewer #3: Low prevalence of Plasmodium falciparum histidine-rich protein 2 and 3 gene deletions in malaria-endemic states of India. 

The authors present a short report on hrp deletions in India across multiple malaria endemic states spanning several years. The sample number analysed is large and the authors find a few suspected deletions. It would have been great to have some more details on the size of the study sites and distances between the sampling locations. 

Response: Thank you very much for your constructive comments, which will undoubtedly enhance the manuscript for a global audience. Regarding the concern about sample size, we would like to clarify that these are old samples used in therapeutic efficacy studies in India from 2014, 2017, 2019, and 2020 (Ref 10, 11). We have also included additional details in accordance with your recommendations.

1. The map (figure 1) has no scale, and if I interpret this correctly, the samples were collected from different locations in different years. 

Response: Figure 1 is a map of India that is self-explanatory. The coloured dots indicate the years in which samples were collected, while the arrows show the districts within the states.

2. As there is no details provided in what malaria transmission is like in the different states and whether it is homogeneous across states, not too many conclusions can be drawn.

Response: Thank you for your feedback. We have incorp

---

## [Decision Letter · Decision Letter 1]

29 Nov 2024

Low prevalence of Plasmodium falciparum histidine-rich protein 2 and 3 gene deletions in malaria-endemic states of India

PONE-D-24-27945R1

Dear Dr. Bharti,

We’re pleased to inform you that your manuscript has been judged scientifically suitable for publication and will be formally accepted for publication once it meets all outstanding technical requirements.

Kind regards,

Henk Schallig, Ph.D

Academic Editor

PLOS ONE

Additional Editor Comments (optional):

Reviewers' comments:

Reviewer's Responses to Questions

**Comments to the Author**

1. If the authors have adequately addressed your comments raised in a previous round of review and you feel that this manuscript is now acceptable for publication, you may indicate that here to bypass the “Comments to the Author” section, enter your conflict of interest statement in the “Confidential to Editor” section, and submit your "Accept" recommendation.

Reviewer #3: All comments have been addressed

2. Is the manuscript technically sound, and do the data support the conclusions?

Reviewer #3: Yes

3. Has the statistical analysis been performed appropriately and rigorously? 

Reviewer #3: Yes

4. Have the authors made all data underlying the findings in their manuscript fully available?

Reviewer #3: Yes

5. Is the manuscript presented in an intelligible fashion and written in standard English?

Reviewer #3: Yes

6. Review Comments to the Author

Reviewer #3: (No Response)

7. PLOS authors have the option to publish the peer review history of their article (what does this mean?). If published, this will include your full peer review and any attached files.

Reviewer #3: No

---

## [Editor Report · Acceptance letter]

3 Dec 2024

PONE-D-24-27945R1 

PLOS ONE

Dear Dr. Bharti, 

I'm pleased to inform you that your manuscript has been deemed suitable for publication in PLOS ONE. Congratulations! Your manuscript is now being handed over to our production team.

Kind regards, 

on behalf of

Dr. Henk Schallig 

Academic Editor

PLOS ONE